# Burden of stigma among tuberculosis patients in a pastoralist community in Kenya: A mixed methods study

**Grace Wambura Mbuthia**[1]*, **Henry D. N. Nyamogoba**[2], **Silvia S. Chiang**[3,4], **Stephen T. McGarvey**[5]

**1** College of Health Sciences, Jomo Kenyatta University of Agriculture and Technology, Nairobi, Kenya,
**2** College of Health Sciences, Moi University, Eldoret, Kenya, **3** Department of Pediatrics, Alpert Medical School, Brown University, Providence, RI, United States of America, **4** Center for International Health Research, Rhode Island Hospital, Providence, RI, United States of America, **5** International Health Institute, and Department of Epidemiology, School of Public Health, Brown University, Providence, RI, United States of America

* grace.mbuthia@jkuat.ac.ke

## Abstract

### Background

Tuberculosis (TB) stigma remains a barrier to early diagnosis and treatment completion. Increased understanding of stigma is necessary for improved interventions to minimise TB stigma and its effects. The purpose of this study is to quantitatively measure TB stigma and to explore qualitatively its manifestation among TB patients in a rural Kenyan community.

### Methods

This hospital based study using explanatory sequential mixed methods approach was conducted in 2016. In the quantitative part of the study, a questionnaire containing socio-demographic characteristics and scales measuring perceived TB stigma and experienced TB stigma, was administered to 208 adult pulmonary TB patients receiving treatment in West Pokot County. Respondents with high stigma were purposively selected to take part in in-depth interviews and focus group discussions. The qualitative data were collected through 15 in-depth interviews and 6 focus group discussions with TB patients. Descriptive and bivariate analysis was done for the quantitative data while the thematic analysis was done for qualitative data.

### Results

The internal consistency reliability coefficients were satisfactory with Cronbach alphas of 0.87 and 0.86 for the 11-item and 12-item stigma measurement scale. The investigation revealed that TB stigma was high. The key drivers of TB stigma were the association of TB with HIV/AIDS and the fear of TB transmission. TB stigma was exemplified through patients being isolated by others, self-isolation, fear to disclose TB diagnosis, association of TB with human immunodeficiency virus (HIV) and lack of social support. Being a woman was

**Data Availability Statement:** In order to keep the privacy of the study participants, the qualitative data will not be shared publicly. Qualitative data from this study are available from the approving

ethics committee (Moi University College of Health Sciences/Moi Teaching and Referral Hospital Institutional Research and Ethics Committee) upon written requests under the ethical provisions of the body. Researchers who meet the criteria for access to confidential data, can write to Board Chair, Institutional Research Ethics Committee (IREC) Moi University College of Health Sciences & Moi Teaching and Referral Hospital, P.O. Box. 3-30100 Eldoret, Kenya (Office line: +254787723677; Email: irecmtrh@gmail.com or contact@irec.or.ke).

**Funding:** This research was supported by the Consortium for Advanced Research Training in Africa (CARTA). CARTA is jointly led by the African Population and Health Research Center and the University of the Witwatersrand and funded by the Carnegie Corporation of New York(Grant No: B 8606 R02), Sida (Grant No: 54100113), the DELTAS Africa Initiative (Grant No: 107768/Z/15/Z), The DELA Africa Initiative is an independent funding scheme of the African Academy of sciences (AAS)'s Alliance for the Accelerating Excellence in Science in Africa(AESA) and supported by the New Partnership for Africa's Development Planning and Coordinating Agency (NEPAD Agency) with funding from Wellcome Trust (UK) and the UK government. The statements made and views expressed are solely the responsibility of the fellow.

**Competing interests:** The authors have declared that no competing interests exist.

significantly associated with high levels of both experienced stigma (p = 0.007) and perceived stigma (p = 0.005) while age, marital status, occupation and the patient's religion were not.

## Conclusion

There is a need to implement stigma reduction interventions in order to improve TB program outcomes.

## Introduction

Tuberculosis (TB) causes an estimated 10 million new cases and 1 million deaths per year [1]. Stigma associated with TB is a barrier to early diagnosis and treatment adherence [2–8] and therefore impedes TB-control efforts. The World Health Organization's End TB Strategy explicitly states that TB-associated stigma needs to be alleviated in order to achieve TB elimination [9].

The term 'stigma' refers to "exclusion, rejection, blame, or devaluation resulting from experience or reasonable anticipation of an adverse social judgment" because of a particular condition [10]. Similarly, stigma refers to a social process in which people, out of fear of the disease, aim to maintain social control by contrasting those who are normal with those who are different [11]. Stigma leads to feelings of shame, tainted identity, increases the stress associated with illness, and contribute to psychological and social morbidity [12]. Due to the fear of being identified as TB patients, individuals may not seek health care services for TB [13], leading to delays in diagnosis, treatment, and increased community transmission [14, 15]. Similarly, TB stigma is a barrier to adherence, as patients may not go to the health centre to receive treatment for fear that their neighbours or friends will see them [16–18].

Historically, TB has been stigmatized because of its contagious nature, ignorance about its cause, transmission, or treatment, as well as its association with marginalized groups [19–23]. Stigma associated with HIV and TB often overlap due to high coinfection rate in some parts of the world [24]. Stigma associated with HIV /AIDS(acquired immunodeficiency syndrome) therefore, increases TB stigma [21, 25] and may further result in delays in seeking care [26, 27].

Although TB stigma is recognised as a serious problem, it has been difficult to describe the magnitude and the public health importance of the problem due to a lack of tools to quantify stigma in the African context. In the absence of an instrument to measure TB stigma in Kenya at the time of this study, we examined the validity and reliability of the TB stigma scales previously developed and validated in Thailand [26].

Disease-related stigma has been viewed as a difficult construct to measure [28–31]. Most studies that measure stigma rely on narrative accounts of prior experiences with stigma [28]. This study sought to use a mixed methods approach to study the issue of TB stigma. The advocates for mixed quantitative and qualitative methods in stigma research believe that the approach is more applicable in studying complex research questions that may be deeply socially contextual compared to the traditional qualitative or quantitative alone studies [32, 33]. This is based on the argument that each method has its own limitations and use of only one method is likely to yield biases and limited results. When used together, the methods complement each other and yield a more robust analysis of the problem under study [34, 35]. In addition, combining the methods may be quite useful for early studies of stigma in societies where little work has been done. The aim of this study is to quantitatively measure TB stigma

and use qualitative methods to explore its manifestation among TB patients in a rural Kenyan community.

## Methods

### Study setting, population and design

This facility-based study was carried out from January to June 2016 in four hospitals offering TB services in West Pokot County, Kenya [36]. The County is located in the Rift Valley. It has a population of 512,690 people (2009 census) and an area of 9,169.4 km$^2$. It is less developed with poor infrastructure, and the inhabitants are mainly ethnic Pokots. Approximately 80% of the County is arid or semiarid, and 60% of the inhabitants are nomadic pastoralists while the rest of the population are agro-pastoralists [37]. There are 21 TB treatment sites in the county. The main government-run treatment sites are one County hospital and three Sub County hospitals included in this study. The rest include faith-based health facilities, health centres and dispensaries offering TB treatment.

West Pokot County had a higher TB case notification rate of 225 per 100,000 population compared to the national rate of 217 per 100,000 population [38]. The HIV prevalence in Kenya is 4.9% with a high TB/HIV co-infection rate of 41.8% in Western Kenya [39].

This study combined quantitative and qualitative methods, leading to a mixed method approach [40]. The level of stigma was determined using stigma scores, while focus group discussions (FGDs) and in-depth interviews (IDIs) were used to understand the patient's experiences of stigma and how TB stigma is manifested in this setting. Selection of participants for the FGDs and IDIs was based on stigma scores from the quantitative phase of the study. The qualitative data was also used to provide detailed explanations to the patterns observed in the quantitative data.

### Sample size and sampling

The study sample was 208 adult patients chosen from a total of 635 new TB cases registered during the study period. Adult pulmonary TB patients currently in the first 2 months of treatment for TB—but who had completed at least 2 weeks of treatment—were included in the study. Patients with mental illness diagnosed before the TB diagnosis and multi-drug resistant TB were excluded from the study.

Every patient seen in the TB clinics of the four participating hospitals during the TB clinic days (normally Tuesdays and Thursdays) who met the inclusion criteria and was willing to take part in the study was included in the study. Patients were distributed equally across the four hospitals.

From the 208 respondents, 61 were purposively selected to take part in the qualitative study component based on scoring high or very high stigma scores. We also ensured both sexes and age categories were well represented among the participants.

### Data collection methods

This study combined quantitative and qualitative methods [40]. A two-stage exploratory sequential mixed method approach was used [41, 42]. The point of integration of the methods was in the selection of participants for the qualitative data collection and in the interpretation of the findings. The first phase included a quantitative survey done using a questionnaire and the second phase, a qualitative part of the study, consisted of FGDs and IDIs.

**Quantitative data: Survey.** The interviewer-administered questionnaire contained items to assess socio-demographic characteristics, and a stigma measurement scale developed and

validated in Thailand to quantify TB associated stigma at individual and community level [26]. The questionnaire was translated to the Kiswahili language and was administered at the TB clinic at the end of their clinic visit by the lead author (GWM) and a local research assistant. The scale contained two subscales: 12 items focused on the respondent's experiences, thoughts and feelings of TB stigma (experienced stigma), and 11 items assessing how the respondent perceived the community to feel about or act toward people with TB (perceived stigma). Stigma items were scored on a Likert scale with four levels: strongly disagree (0), disagree (1), agree (2), and strongly agree (3), and responses were summed for each scale to create stigma scores, with higher responses indicating higher stigma.

**Qualitative data: In-depth interviews.** The in-depth interviews took 60–90 minutes and were conducted at the TB clinics to collect data on patient experiences of stigma during their illness following a semi-structured interview guide. The interview guide included questions such as; "How did you interact with the rest of the family /community after realizing you had TB?", "How did those around you treat you when they knew you had TB? (Probes: social support system from family, friends, and the community), "Did you experience any form of discrimination/stigma before, during and after treatment? How? Why?".

To avoid the power distance of the provider on patient response, the interview was conducted after the patient had already been served by the health worker so as not to make them feel obliged to answer the questions. The interview was also conducted in a separate private room in the absence of the health care provider. GWM a PhD student in medical anthropology and a Pokot-speaking research assistant moderated the interviews and recorded notes during the interviews. The interviews were conducted in Kiswahili. The Kiswahili audio-recorded interviews were translated into English and transcribed in English.

**Focus group discussion.** Focus group discussions occurred at the TB clinics, ranged from 6–10 respondents. Males and females FGDs were done separately. Each FGD lasted 60 to 90minutes. A semi-structured FGD guide was used to collect data. The guide included questions such as; "Do TB patients face any form of discrimination?" "How do TB patients interact with the rest of the family /community (probes: do they isolate themselves or are they isolated by others? how?" "How does the family/community members expect TB patients to conduct themselves?" "What social support do TB patients receive from family/community?"

GWM moderated the discussion, and the research assistant recorded notes during the discussion which were conducted in Kiswahili language. The interviews were conducted until theoretical saturation was reached, meaning no new conceptual information was emerging from further discussion [43]. Data saturation was reached at 6 FGDs. Each FGD was audio-recorded and later transcribed verbatim [44]. The Kiswahili audio-recorded discussion were translated into English and transcribed in English.

## Data analysis

The quantitative data were analysed using Stata software v13.1 (College Station, Texas, US). To test the adaptability of a TB stigma measurement scale to our setting, we conducted exploratory factor analysis and calculated Cronbach's alphas to determine the internal consistency. A scale with an alpha of $\geq 0.7$ was considered to have good reliability. Standard techniques for analyses of univariate descriptions and bivariate associations were used. Statistical significance was set at $P < 0.05$.

The transcribed transcripts were analysed with NVivo v11 (QSR International Melbourne, Australia). Thematic analysis was done by reading through the transcript multiple times and identifying, coding, and categorizing meaningful patterns into themes and subthemes [45]. GWM coded all the transcripts and together with the research assistant identified emerging

themes while two additional authors (HDN and SSC) offered guidance. Data were presented in textual and where possible, verbatim quotes used to amplify the voices of the respondents.

## Ethical considerations

The research was approved by Moi University School of Medicine / Moi Teaching and Referral Hospital Institutional Research and Ethics Committee (Formal Approval Number: IREC 0001349). Participants provided written informed consent in Kiswahili, which is commonly spoken by the majority of the West Pokot residents, prior to data collection.

# Results

## Background information for participants

Of the 208 respondents who took part in the survey, 132 (63.5%) were male. The age range of the participants was 18–78 years with median age of 36 (IQR 25–50) years. Eighty-one (81%) of the respondents identified as Christian, 49 (23.5%) of the respondents had attained secondary school education, and 104 (50%) were farmers (Table 1). Sixty-one (61%) participated in the qualitative portion of the study; 15 (7 women; 8 men) participated in IDIs, and 46 (22 women; 24 men) participated in FGDs. Age of the 61 participants ranged from 27–61 years.

## Tuberculosis stigma scale characteristics

The Cronbach alpha internal consistency reliability coefficient was used to determine the validity of the TB stigma scales. The internal consistency for the two scales indicated by the

**Table 1. Socio-demographic characteristics of 208 TB patients in West Pokot County.**

| Variables | Male n = 132 n (%) | Female n = 76 n (%) | Total n = 208 n (%) |
|---|---|---|---|
| **Age group (years)** | | | |
| 18–20 | 17(12.9) | 13(17.1) | 30(14.4) |
| 21–30 | 29(22) | 21(27.6) | 50(24) |
| 31–40 | 29(22) | 16(21.1) | 45(21.6) |
| 41–50 | 23(17.4) | 14(18.4) | 37(17.9) |
| >50 | 34(26.7) | 12(15.8) | 46(22.1) |
| **Religion** | | | |
| Christianity | 103(78) | 66(86.8) | 169(81.3) |
| Islam | 8(6) | 4(5.3) | 12(5.7) |
| No religion | 21(16) | 6(7.9) | 27(13) |
| **Level of education** | | | |
| No education | 60(45.5) | 25(32.9) | 85(40.9) |
| Primary | 44(33.3) | 30(39.5) | 74(35.6) |
| Secondary | 28(21.2) | 21(27.6) | 49(23.5) |
| **Marital status** | | | |
| Married | 85(64.4) | 40(52.6) | 125(60.1) |
| Single | 32(24.2) | 24(31.6) | 56(26.9) |
| Divorced/widowed | 15(11.4) | 12(15.8) | 27(13) |
| **Occupation** | | | |
| Farmer | 69(52.3) | 35(46.1) | 104 (50) |
| Businessman | 25(18.9) | 15(19.7) | 40(19.2) |
| Formal employment | 23(17.4) | 14(18.4) | 37(17.8) |
| Others | 15(11.4) | 12(15.8) | 27(13) |

**Table 2. Stigma scale characteristics.**

| Tuberculosis stigma scale | Cronbach's Alpha | Mean stigma score (SD) | Minimum–Maximum score |
|---|---|---|---|
| Experienced TB stigma (12 items) | 0.86 | 20.67 (5.78) | 5–33 |
| Perceived TB stigma (11 items) | 0.87 | 17.03 (5.34) | 6–36 |

Cronbach's coefficient alpha of 0.87 and 0.86 for the 11-item and 12-item scale was satisfactory. The summed stigma scores were normally distributed with mean scores of 20.7 for the experienced and 17 for the perceived TB stigma (Table 2).

## Tuberculosis stigma scale loading value and scores

Factor analysis showed high loadings (>0.40) for all the items in both the perceived stigma scale and the experienced stigma scale (Table 3). Items with absolute loading value of ≥ 0.40 are considered important in the scale [46].

Items that were rated highest in the experienced stigma scale were those that pertained to the association of TB and HIV/AIDs. Similarly, items covering fear of disclosure and getting TB through casual interactions were rated high.

**Table 3. Absolute loading values for TB stigma scales by item.**

| A. Community perspectives on tuberculosis (assessed by patients) | Mean | Loading value |
|---|---|---|
| i. Some people may not want to eat or drink with friends who have TB | 1.70 | 0.51 |
| ii. Some people feel uncomfortable about being near those with TB | 1.72 | 0.68 |
| iii. If a person has TB, some community members will behave differently towards that person for the rest of his/her life | 1.45 | 0.61 |
| iv. Some people do not want those with TB playing with their children | 1.49 | 0.67 |
| v. Some people keep their distance from people with TB | 1.59 | 0.69 |
| vi. Some people think that those with TB are disgusting | 1.35 | 0.67 |
| vii. Some people do not want to talk to others with TB | 1.35 | 0.70 |
| viii. Some people are afraid of those with TB | 1.5 | 0.66 |
| ix. Some people try not to touch others with TB | 1.44 | 0.72 |
| x. Some people may not want to eat or drink with relatives who have TB | 1.50 | 0.69 |
| xi. Some people prefer not to have those with TB living in their community | 1.60 | 0.66 |
| **B. Perspectives of TB patients on tuberculosis (experienced stigma)** | | |
| i. Some people who have TB feel hurt of how others react to knowing they have TB | 1.66 | 0.60 |
| ii. Some people who have TB lose friends when they share with them they have TB | 1.61 | 0.68 |
| iii. Some people who have TB feel alone | 1.61 | 0.76 |
| iv. Some people who have TB keep their distance from others to avoid spreading TB germs | 1.70 | 0.51 |
| v. Some people who have TB are afraid to tell those outside their family that they have TB | 1.78 | 0.65 |
| vi. Some people who have TB are afraid of going to TB clinics because other people may see them there | 1.33 | 0.61 |
| vii. Some people who have TB are afraid to tell others that they have TB because others may think that they also have AIDS | 2.10 | 0.71 |
| viii. Some people who have TB feel guilty because their family has the burden of caring for them | 1.50 | 0.61 |
| ix. Some people who have TB will choose carefully who they tell about having TB. | 1.92 | 0.63 |
| x. Some people who have TB feel guilty for getting TB because of their smoking, drinking, or other careless behaviours. | 1.61 | 0.45 |
| xi. Some people who have TB are worried about having AIDS | 1.89 | 0.67 |
| xii. Some people who have TB are afraid to tell their family that they have TB | 1.32 | 0.55 |

**Table 4. Level of experienced and perceived stigma among 208 TB patients in West Pokot County.**

| Level of experienced stigma | Frequency (%) | Level of perceived stigma | Frequency (%) |
|---|---|---|---|
| Low stigma (Score 1–12) | 27(12.9) | Low stigma (Score 1–11) | 55 (26.4) |
| High stigma (Score of 13–24) | 127(61.1) | High stigma (Score of 12–22) | 121(58.2) |
| Very high stigma (Score of 25–36) | 54(25.9) | Very high stigma (Score of 23–33) | 32 (15.4) |

## Level of stigma

The patient's experienced stigma scores were classified into three categories, and the majority, (61.1%) had a high level of experienced stigma. Similarly, patients' perceived stigma scores were grouped into three categories and 58.2% had a high level of perceived stigma (Table 4).

## Reasons underlying stigma

The survey showed that being female was significantly associated with high levels of both experienced stigma (p = 0.007) and perceived stigma (p = 0.005). Twenty-nine (54%) females reported very high level of experienced stigma, compared to 25 (46%) males. Twenty (57.2%) females, compared to 15 (42.8%) males, reported very high level of perceived stigma. The age, religion, level of education, marital status and occupation did not differ significantly with both the level of perceived and experienced stigma.

Data from both the IDIs and focus group discussions revealed that the key drivers of TB stigma were the association of TB with HIV/AIDS and the fear of TB transmission. This resulted in shunning and discrimination of TB patients.

**Association of TB with HIV.** The participants reported that when one has TB they are considered to have HIV also, and therefore face discrimination. According to the respondents, gaining weight after TB treatment was attributed to the use of antiretroviral (ARV) therapy.

One respondent had this to say;

*"But when I told people I had TB they did not believe me they said that I had HIV. And when I started my TB treatment and had gained weight they said I was taking ARVs and the drugs had made me to gain weight".* (Female 49 years)

The community accepts TB as a disease that occurs due to natural causes and thus do not blame the patient. Conversely, an HIV patient is blamed for having acquired the infection. Therefore, stigma associated with HIV compounds TB stigma. One respondent alluded to this;

*"The reason why they treat you like that is because they thought because I had TB, I had HIV as well and they would say it was my fault, I brought the disease to myself".* (Female FGD one).

Due to the stigma attached to HIV, some of the respondents found it difficult to refer to HIV/AIDS by its name and would use terms such as *"the big disease", "the other one", "the other bad disease".* Respondents noted,

*"Yes that is true when one has TB, people just conclude you have **the other one.**"* (Male FGD one)

*"My mother would tell me, my daughter nowadays they say there are drugs, why don't you test for **the big disease.**"* (Female 38 years).

Respondents reported their preference to keep secret their TB diagnosis due to fear that disclosing their diagnosis would lead to community members' suspicion of having HIV/AIDS. Two female participants said the following;

*"When you have TB you don't want to tell everybody because when they know they start suspecting that you have the other bad disease."* (Female 30 years).

*"but when you have TB people think that you have HIV as well and so we don't go telling everybody that you have TB because they will say you have AIDs as well".* (Female 39 years)

**Fear of transmission.** Respondents reported that community members shun them because they saw them as contagious. Having TB made it impossible to socialise with others and even attend community gatherings. It was distressing to patients to discover that the people whom they counted on for support ostracised them as well. Participants painfully recounted the experience they had to go through;

*". . .. even now no one wants to get close to me. My sisters isolated me and everyone who used to come and laugh with me and eat with me were no more no one wanted to be close to me. They left me. . . I suffered a lot."* (Female 38 years).

*"When I learnt that I had TB, I felt very bad. People could see me and run away. . .When there were wazee Barazas (informal meetings for men) I was barred from attending so as not to infect others."* (Male 42 years).

*"Throughout the period I was sick, it's only my family that remained close to me. . . but most friends abandoned me and I had less and less company."* (Male 28 years).

In addition to being isolated respondents also reported isolating themselves for fear of infecting their loved ones. Others avoided interacting with the rest of the community so as not to be blamed for infecting them. This was illustrated as follows;

*". . ... most of the time you will find me alone at home I prefer to stay home with my disease so that they don't say I passed the disease to other people."* (Female 30 years).

*"It is important to avoid mixing with the rest of the family so as not to infect them. In my case I sent the children to their grandmother so that they don't get infected. They have stayed there for one year now."* (Male FGD two).

*"This disease can be transmitted when one coughs. And so I decided to stay far from my family so that I wouldn't infect them. For the entire period I was on treatment I used to live in a separate house from that of my wife. I did not want to infect her."* (Male 42 years).

The isolation due to fear of TB transmission was done for prolonged period time. For some the isolation would span till the completion of TB treatment and for others it went beyond the treatment period.

*"Personally, I have witnessed this, I have been Isolated. I sleep in my own house and have my own utensils until I complete my eight months of treatment".* (Female FGD 1)

*"Until one finish the medication and one should be isolated for 6 months until the person has finished taking the medication".* (Female FGD 3)

*"In my case from the time I suspected TB, I sent the children to their grandmother so that they don't get infected. They have stayed there for one year now. It is for their own good. My wife remained with me but we don't sleep in the same bed. I don't want to infect her".* (Male FGD 2)

The female patients verbalised having been isolated more than the male patients. The men expressed a greater sense of belonging and therefore able to resist isolation. Most men reported that their family was supportive while for many women discrimination came from both family and the rest of the community.

*"I got very little help from friends. I was isolated from friends and families, when they got to know that I had TB, no one came near me". (*Female 36 years)

*"The neighbours stopped coming to my home and were only talking ill about me".* (Female 49 years).

When asked whether he faced any kind of isolation or discrimination, one male respondent had the following to say: "*I live in my own home why would anyone isolate me".* (Male 45 years).

Social support was key in overcoming TB stigma. Respondents emphasized the importance of social support during their illness and recovery. While the neighbours and in some cases friends abandoned the patient, the unconditional love of family members remained a key source of social support for some. The love of the family members encouraged patients to complete their treatment.

*"Throughout the period I was sick, it's only my family that remained close to me. . . but most friends abandoned me and I had less and less company. My family helped me to overcome the illness."* (Male 28 years).

## Discussion

The study showed high levels of both perceived and experienced stigma among TB patients which was mainly driven by the association of TB with HIV/AIDS and the fear of TB transmission. To the best of our knowledge, this is the first study to validate a TB stigma measurement scale developed elsewhere among TB patients in Kenya. We found that the stigma measurement scale developed in Thailand and adopted in this research study had good overall internal consistency, reliability, and psychometric characteristics among TB patients in West Pokot County, Kenya.

Understanding the source of TB stigma is integral to reducing its effect on health seeking behaviour that often leads to delayed diagnosis. Our study showed that fear of infection through casual transmission was a significant determinant of TB related stigma. Items in the stigma scale linked to fear of infection through casual interaction, such as eating or drinking with relatives rated highly on the Likert scale. The community's understanding of the contagious nature of TB often led patients to consider themselves disease vectors and as a result they self- isolated. One participant mentioned, '*some people who have tuberculosis keep their distance from others to avoid spreading TB germs'.* Similarly, from the interviews the respondents confirmed that they faced social isolation from community members but also from friends and family in severe cases. Self-stigmatisation was clearly evident from the study results. The respondents self-isolated themselves in fear of infecting their loved ones particularly children and to avoid being shunned or being the subject of gossip by neighbours. Cases of TB patients

sending their children to live with the grandparents as well as avoidance of sharing the same house with a spouse were common.

Social isolation of tuberculosis patients is common and dates back to the 18[th] century where those considered to be in advanced stages of tuberculosis were not attended to by health professionals but were instead sent to the temple and sea voyage for cure. While these were delivered as form of treatment, the real intention was to isolate people from the rest of the population to prevent disease transmission [47]. Evidence has shown that perceived risk of transmitting TB to the healthy community members is a leading cause of stigmatization in other settings [2, 14, 31, 48–50]. The current study shows that, in an attempt to deal with the risk of infecting others, the patients avoided sharing meals and kitchen dishes, sharing a room, and having sexual relationships. Due to lack of information among the community members, the fear of infection often is exaggerated and usually commences after the diagnosis and may persist even after completion of treatment. However, this kind of stigma is due to incorrect knowledge about transmission of TB. Although there is great risk of TB transmission when one has prolonged close contact with untreated acid-fast smear positive TB patients, in general, after two weeks of effective treatment, most patients are unlikely to transmit the disease [51].

Both the qualitative and quantitative data show that TB stigma manifested in patients as a fear of disclosure of their disease status. In the present study, maintaining confidentiality of a TB diagnosis was rated high. This is likely to affect TB control since a fear of being identified as a TB patients, may prevent an individual from seeking health care services for a TB diagnosis [13]; this in turn would lead to delays in a diagnosis and therefore, increased transmission of the disease within a community [14, 15].

Similar to other studies [24, 26, 31, 52], the current study found that the association of TB and HIV infection/AIDS increases TB stigma. Items such as 'Some people who have TB are worried about having AIDS" were rated highest. Similarly, the IDIs confirmed that patients with TB feared disclosing their diagnosis so as not to be labelled as HIV positive. In Sub Sahara Africa where the prevalence of HIV infection/AIDS is high, a patient with TB is often assumed to have the virus. Tuberculosis has often been perceived as a surrogate marker for HIV positivity [27]. This is mainly due to the high coinfection rate of HIV and TB as well as the similarity of symptoms such as loss of weight. A previous study done in western Kenya showed a high rate of TB HIV/AIDs coinfection rate of 42% higher than the national coinfection rate of 39% [39].

The survey showed that being female was associated with higher stigma scores. Similarly, in the IDIs and FGDs more female patients reported having faced discrimination as result of their illness compared to their male counterparts. This could be attributed to the fact that males were in a better position to resist discrimination particularly in the family setup due to a greater sense of belonging and community status. In rural parts of Kenya, the man is seen as the head of the family and gender inequalities exist with women lacking the ability to own land. The male patients benefited from the caring nature of women and as result did not experience discrimination at the family level.

While the survey showed that some people who have TB blame themselves for getting TB because of their smoking, drinking, or other behaviours, this finding was not seen in the IDIs and FGDs. This difference may be attributed to the discomfort of admitting one's guilt during the discussions or lack of probing on association of TB and these behaviours. Studies have shown that alcohol abuse, poverty, low education level are not only important social determinants of TB onset but also significant predictors of therapy failure and MDR in people with TB [53, 54]. The strong association between health and inequality requires an acceleration of social and innovative health progress among poor and socially excluded groups for the success of TB control in TB/HIV high burden settings [53, 55].

The strength of this study rests in the use of a mixed methods approach to assess TB related stigma. The survey allowed us to measure the level of stigma while the qualitative part of the study allowed for exploration of patients' perceptions and experiences of TB stigma.

However, the study has limitations in that it focuses on individuals affected by TB only and not the community as a whole. Therefore, we could only document the affected individuals' self-perceived manifestations of stigma related to TB but not the views of the general community.

## Conclusion

The study reveals high levels of both perceived and experienced stigma among TB patients in the pastoralist community in Kenya. There is therefore a need to design effective strategies to reduce social stigma and its effects among marginalized populations. In addition, there is a need to engage the affected communities to address stigma through support groups and health education to demystify misconceptions surrounding TB and in turn reduce stigma. There is also a need for further studies on the community's knowledge of TB transmission and how this drives TB stigma in the community.

## Supporting information

**S1 File. Interview guide.**
(DOCX)

**S1 Data. Survey data set.**
(DTA)

## Acknowledgments

We are thankful to the West Pokot County and sub county TB/Leprosy coordinators for their assistant during the data collection. We also thank the patients who accepted to take part in this study. We are also indebted to Dr. Daria Szkwarko and Dr. Jane Carter for reviewing and editing this paper.

## Author Contributions

**Conceptualization:** Grace Wambura Mbuthia, Henry D. N. Nyamogoba.

**Data curation:** Grace Wambura Mbuthia.

**Formal analysis:** Grace Wambura Mbuthia.

**Funding acquisition:** Grace Wambura Mbuthia.

**Investigation:** Grace Wambura Mbuthia.

**Methodology:** Grace Wambura Mbuthia.

**Resources:** Henry D. N. Nyamogoba.

**Supervision:** Henry D. N. Nyamogoba, Silvia S. Chiang, Stephen T. McGarvey.

**Visualization:** Grace Wambura Mbuthia.

**Writing – original draft:** Grace Wambura Mbuthia.

**Writing – review & editing:** Grace Wambura Mbuthia, Henry D. N. Nyamogoba, Silvia S. Chiang, Stephen T. McGarvey.

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
