## [Decision Letter · Decision Letter 0]

25 Aug 2020

PONE-D-20-12181

Burden of stigma among tuberculosis patients in a pastoralist community in Kenya: A mixed methods study

PLOS ONE

Dear Dr.ssa Grace Wambura Mbuthia,

Thank you for submitting your manuscript to PLOS ONE. After careful consideration, we feel that it has merit but does not fully meet PLOS ONE’s publication criteria as it currently stands. Therefore, we invite you to submit a revised version of the manuscript that addresses the points raised during the review process.

We look forward to receiving your revised manuscript.

Kind regards,

Claudia Marotta

Academic Editor

PLOS ONE

Journal Requirements:

2. Please include additional information regarding the survey or questionnaire used in the study and ensure that you have provided sufficient details that others could replicate the analyses. For instance, if you developed a questionnaire as part of this study and it is not under a copyright more restrictive than CC-BY, please include a copy, in both the original language and English, as Supporting Information. Moreover, please include more details on how the questionnaire was pre-tested, and whether it was validated.

3.We note that you have indicated that data from this study are available upon request. PLOS only allows data to be available upon request if there are legal or ethical restrictions on sharing data publicly. For information on unacceptable data access restrictions, please see http://journals.plos.org/plosone/s/data-availability#loc-unacceptable-data-access-restrictions.

Additional Editor Comments (if provided):

Dear Authors

Below reviewers suggestions

Reviewers' comments:

Reviewer's Responses to Questions

**Comments to the Author**

1. Is the manuscript technically sound, and do the data support the conclusions?

Reviewer #1: Yes

Reviewer #2: Yes

Reviewer #3: Partly

2. Has the statistical analysis been performed appropriately and rigorously? 

Reviewer #1: Yes

Reviewer #2: Yes

Reviewer #3: N/A

3. Have the authors made all data underlying the findings in their manuscript fully available?

Reviewer #1: Yes

Reviewer #2: Yes

Reviewer #3: Yes

4. Is the manuscript presented in an intelligible fashion and written in standard English?

Reviewer #1: Yes

Reviewer #2: No

Reviewer #3: Yes

5. Review Comments to the Author

Reviewer #1: Authors wrote a very interesting manuscript on an important issue. Congratulations.

Only some suggestions:

1. Abstract: please improve abstract

1.Introduction: delete line 58

2. Methods: specifies the period and type of the study

Delete line 172

3. Results, Figure and Tables: I appreciate a lot, delete line 184,202-203, 206-208

4. Discussion and Conclusion: if you can improve your discussion with some items:

- the role of social determinants oh health in tb

(Di Gennaro F, Pizzol D, Cebola B, et al. Social determinants of therapy failure and multi drug resistance among people with tuberculosis: A review. Tuberculosis (Edinb). 2017;103:44‐51.) and the more attention that need tb and hiv patients also in stigma issue in high burden setting of TB (Bobbio F, al. Focused ultrasound to diagnose HIV-associated tuberculosis (FASH) in the extremely resource-limited setting of South Sudan: a cross-sectional study. BMJ Open. 2019;9(4):e027179) and how stigma and SDH are predictors of treatment failure (Pizzol D, Veronese N, Marotta C, et al. Predictors of therapy failure in newly diagnosed pulmonary tuberculosis cases in Beira, Mozambique. BMC Res Notes. 2018;11(1):99)

Reviewer #2: Many thanks to the authors of these paper for taking time to examine a very important issue around tuberculosis experienced and perceived stigma in a pastoralist community. The strength of the study lies in its sequential exploratory mixed approach that combines quantitative and qualitative designs. The authors finding of how gender influences perceived and experience TB stigma has many implications for policy and practices.

General comments

Introduction

Author was able to review previous literature, identify gaps in knowledge and justification for the study

Methodology: Even though a purposive sampling was used, can the author explain how they know the study is well powered to support the conclusions? How did the researchers achieve FGD saturation?In what language was the analysis done since the interview was in Kiswali any back translation?

The IDI took place in the clinic where TB providers are domiciled.How did the authors avoid power distance of providers on patient response?

Results

Tables 3 ande 4 can be combined since it’s the same scale. The questions should be numbered

Tables 5 and 6 can also be combined into one

Discussion

The author should highlight important significant findings first. While it is commendable how the authors highlighted commonalities in the findings from the questionnaire and the qualitative (FGD and IDI) such the issue of HIV/AIDS and significant impact of TB stigma on women, they fail to discuss how some of the findings in the 2 designs differ. It will be important to also highlight any differences in the findings from the two designs and any explanations for these. Also this study was conducted in a marginalized community. How does this finding provide unique knowledge in this regards. Any differences in this population and those previously studied?

Conclusion should restate significant findings and future studies. On area I think should be explored is the understanding of the community on TB transmission.Do they know TB patients remain non-infectious after 2 weeks of treatment? And how does this knowledge or its absence drive stigma.

Specific Comments

Page 2 Line 46, states not stages

Page 2 Line 50—uses a different citation site from rest of the paper (page number)

Page 3, Line 61-62 Define HIV/AIDS before the use of the acronym

Page 4, Line 78 Remove ‘and’ between ‘be’ and ‘quite’

‘Of’ not ‘if’ after study

Page 5, Line 100-101 provide further explanation on patient selection for FDG and IDI, why were only those with high stigma scores selected for the FGD/IDI?

Age 5, Line 119-120. You mentioned 4 FGD although you earlier stated 6 FGDs

Line 127, what is the full meaning of GWM?

Reviewer #3: Since, it is important to address stigma related to TB for improved program performance and TB indicators. Hence, the findings of the study reveals important avenues for addressing the health seeking behavior. The stigma measurement scale used in this research is useful. Mainly qualitative in nature, the study opens several reveals the sources of TB stigma that often lead to delayed TB diagnosis.

6. PLOS authors have the option to publish the peer review history of their article (what does this mean?). If published, this will include your full peer review and any attached files.

Reviewer #1: **Yes: **Francesco Di Gennaro

Reviewer #2: No

Reviewer #3: No

---

## [Author Response · Author response to Decision Letter 0]

5 Sep 2020

Dear Reviewers,

Thank you for taking time to review our work. The following is a point by point response to the comments raised.

Reviewer #1: Authors wrote a very interesting manuscript on an important issue. Congratulations. 

Authors’ Reply: Thank you for the compliment.

Abstract: please improve abstract

Authors reply: We have revised the abstract.

1.Introduction: delete line 58 – 

Authors’ Reply: Line 58 has been deleted.

2. Methods: specifies the period and type of the study 

Author Reply: The period and type of study has been stated.

Delete line 172- 

Authors’ Reply: Line 172 has been deleted.

3. Results, Figure and Tables: I appreciate a lot, delete line 184,202-203, 206-208

Authors’ Reply: the lines have been deleted.

4. Discussion and Conclusion: if you can improve your discussion with some items:

- the role of social determinants of health in TB.

(Di Gennaro F, Pizzol D, Cebola B, et al. Social determinants of therapy failure and multi drug resistance among people with tuberculosis: A review. Tuberculosis (Edinb). 2017;103:44‐51.) and the more attention that need tb and hiv patients also in stigma issue in high burden setting of TB (Bobbio F, al. Focused ultrasound to diagnose HIV-associated tuberculosis (FASH) in the extremely resource-limited setting of South Sudan: a cross-sectional study. BMJ Open. 2019;9(4):e027179) and how stigma and SDH are predictors of treatment failure (Pizzol D, Veronese N, Marotta C, et al. Predictors of therapy failure in newly diagnosed pulmonary tuberculosis cases in Beira, Mozambique. BMC Res Notes. 2018;11(1):99)

Authors’ Reply: Many thanks for the suggested papers. We have used and cited them to improve the discussion.

Reviewer #2: Many thanks to the authors of these paper for taking time to examine a very important issue around tuberculosis experienced and perceived stigma in a pastoralist community. The strength of the study lies in its sequential exploratory mixed approach that combines quantitative and qualitative designs. The authors finding of how gender influences perceived and experience TB stigma has many implications for policy and practices. 

Authors’ Reply: We also thank you for taking time to review the work and appreciate your compliments .

Introduction: Author was able to review previous literature, identify gaps in knowledge and justification for the study- 

Authors’ Reply: Thank you for recognizing these sections. 

Methodology: Even though a purposive sampling was used, can the author explain how they know the study is well powered to support the conclusions? How did the researchers achieve FGD saturation? 

Authors’ Reply: This has been clarified. The interviews were conducted until theoretical saturation was reached, meaning no new conceptual information was emerging from further discussion. Data saturation was reached at 6 FGDs. 

In what language was the analysis done since the interview was in Kiswahili any back translation? 

Authors’ Reply: The analysis was done in English. We have indicated that the Kiswahili audio-recorded interviews were translated to English language and transcribed in English language.

The IDI took place in the clinic where TB providers are domiciled. How did the authors avoid power distance of providers on patient response? 

Authors’ Reply: This has been explained in the text as follows; to avoid the power distance of the provider on patient response, the interview was done after the client had already been served by the health worker so as not to make the client feel obliged to answer the questions. The interview was also done in separate private room in the absence of the health care provider

Results

Tables 3 and 4 can be combined since it’s the same scale. The questions should be numbered- 

Authors’ reply: Thank you for this suggestion. The tables have been combined accordingly

Tables 5 and 6 can also be combined into one- 

Authors’ Reply: This has been done. Thank you again!

Discussion

The author should highlight important significant findings first. While it is commendable how the authors highlighted commonalities in the findings from the questionnaire and the qualitative (FGD and IDI) such the issue of HIV/AIDS and significant impact of TB stigma on women, they fail to discuss how some of the findings in the 2 designs differ. It will be important to also highlight any differences in the findings from the two designs and any explanations for these. Also this study was conducted in a marginalized community. How does this finding provide unique knowledge in this regards. Any differences in this population and those previously studied?

Authors’ Reply: The key findings have been highlighted in the first sentence of the discussion. We have also highlighted some of the differences in the quantitative and qualitative findings. A key difference was that quantitative findings revealed some patients feel guilty of causing TB infection due to their behaviors such as smoking or alcohol consumption however this was not reported in the quantitative findings. We attributed this to the fact that, it is more difficult for the respondent to admit such guilt in a one on one discussion or lacked to probe on the same. We also note we did not find unique knowledge on the issue of TB stigma among this marginalized population as the findings compared with those done in different settings.

Conclusion should restate significant findings and future studies. One area I think should be explored is the understanding of the community on TB transmission. Do they know TB patients remain non-infectious after 2 weeks of treatment? And how does this knowledge or its absence drive stigma. 

Authors’ Reply: Thank you for your views and suggestion the need for further studies to understand community knowledge on TB transmission has been added accordingly.

Page 2 Line 46, states not stages- 

Authors’ Reply: This has been corrected 

Page 2 Line 50—uses a different citation site from rest of the paper (page number) –Authors’ Reply: The page number has been removed.

Page 3, Line 61-62 Define HIV/AIDS before the use of the acronym-

Authors’ Reply: The acronym has now been defined.

Page 4, Line 78 Remove ‘and’ between ‘be’ and ‘quite’ –

Authors’ Reply: The word and has been removed

‘Of’ not ‘if’ after study –

Authors’ Reply: This has been corrected

Page 5, Line 100-101 provide further explanation on patient selection for FGD and IDI, why were only those with high stigma scores selected for the FGD/IDI? 

Authors’ Reply: The aim of the qualitative part of the study was to understand the patient’s experiences of stigma and how TB stigma is manifested and for this reason we purposively selected those with substantial level of stigma so as to get deeper understanding of the experience and manifestation of stigma.

Page 5, Line 119-120. You mentioned 4 FGD although you earlier stated 6 FGDs- 

Authors’ Reply: This has been corrected there were 6 FGDs and not four

Line 127, what is the full meaning of GWM? 

Authors’ Reply: These are the initial for the names of the first author- Grace Wambura Mbuthia. For clarity we have defined GWM as lead author.

Reviewer #3: Since, it is important to address stigma related to TB for improved program performance and TB indicators. Hence, the findings of the study reveals important avenues for addressing the health seeking behavior. The stigma measurement scale used in this research is useful. Mainly qualitative in nature, the study opens several reveals the sources of TB stigma that often lead to delayed TB diagnosis. 

Authors’ Reply: We thank you for this compliment

Yours sincerely, 

Grace Mbuthia

Corresponding author

---

## [Decision Letter · Decision Letter 1]

28 Sep 2020

Burden of stigma among tuberculosis patients in a pastoralist community in Kenya: A mixed methods study

PONE-D-20-12181R1

Dear Dr. Mbuthia,

We’re pleased to inform you that your manuscript has been judged scientifically suitable for publication and will be formally accepted for publication once it meets all outstanding technical requirements.

Kind regards,

Claudia Marotta

Academic Editor

PLOS ONE

Additional Editor Comments (optional):

Dear authors, congratulations!

Your paper can be now publish

Reviewers' comments:

Reviewer's Responses to Questions

**Comments to the Author**

1. If the authors have adequately addressed your comments raised in a previous round of review and you feel that this manuscript is now acceptable for publication, you may indicate that here to bypass the “Comments to the Author” section, enter your conflict of interest statement in the “Confidential to Editor” section, and submit your "Accept" recommendation.

Reviewer #1: All comments have been addressed

Reviewer #2: All comments have been addressed

2. Is the manuscript technically sound, and do the data support the conclusions?

Reviewer #1: Yes

Reviewer #2: Yes

3. Has the statistical analysis been performed appropriately and rigorously? 

Reviewer #1: Yes

Reviewer #2: Yes

4. Have the authors made all data underlying the findings in their manuscript fully available?

Reviewer #1: Yes

Reviewer #2: Yes

5. Is the manuscript presented in an intelligible fashion and written in standard English?

Reviewer #1: Yes

Reviewer #2: No

6. Review Comments to the Author

Reviewer #1: Authors improved their manuscript.

I appreciate a lot the job done and the quality and the topic of article.

Well, congratulations!

Reviewer #2: The authors have addressed specific issues raised in the revision. The manuscript is ok in terms of content but will require extensive language editing and formatting

7. PLOS authors have the option to publish the peer review history of their article (what does this mean?). If published, this will include your full peer review and any attached files.

Reviewer #1: No

Reviewer #2: **Yes: **DR ADEPOJU VICTOR ABIOLA

---

## [Editor Report · Acceptance letter]

6 Oct 2020

PONE-D-20-12181R1 

Burden of stigma among tuberculosis patients in a pastoralist community in Kenya: A mixed methods study. 

Dear Dr. Mbuthia:

I'm pleased to inform you that your manuscript has been deemed suitable for publication in PLOS ONE. Congratulations! Your manuscript is now with our production department. 

Kind regards, 

on behalf of

Dr. Claudia Marotta 

Academic Editor

PLOS ONE